# Non-Coding RNAs as Cancer Hallmarks in Chronic Lymphocytic Leukemia

**DOI:** 10.3390/ijms21186720

**Published:** 2020-09-14

**Authors:** Linda Fabris, Jaroslav Juracek, George Calin

**Affiliations:** 1Department of Translational Molecular Pathology, The University of Texas MD Anderson Cancer Center, Houston, TX 77030, USA; lfabris@mdanderson.org; 2Department of Experimental Therapeutics, The University of Texas MD Anderson Cancer Center, Houston, TX 77030, USA; juracekjaroslav@gmail.com

**Keywords:** chronic lymphocytic leukemia, lncRNA, miRNA, hallmarks

## Abstract

The discovery of non-coding RNAs (ncRNAs) and their role in tumor onset and progression has revolutionized the way scientists and clinicians study cancers. This discovery opened new layers of complexity in understanding the fine-tuned regulation of cellular processes leading to cancer. NcRNAs represent a heterogeneous group of transcripts, ranging from a few base pairs to several kilobases, that are able to regulate gene networks and intracellular pathways by interacting with DNA, transcripts or proteins. Deregulation of ncRNAs impinge on several cellular responses and can play a major role in each single hallmark of cancer. This review will focus on the most important short and long non-coding RNAs in chronic lymphocytic leukemia (CLL), highlighting their implications as potential biomarkers and therapeutic targets as they relate to the well-established hallmarks of cancer. The key molecular events in the onset of CLL will be contextualized, taking into account the role of the “dark matter” of the genome.

## 1. Introduction: Current Status of Chronic Lymphocytic Leukemia Research

Chronic lymphocytic leukemia (CLL) is the most common leukemia in adults in the Western world, representing more than 30% of all leukemia cases [1]. It is estimated that 21,040 new CLL cases will be diagnosed in 2020 in the United States [1]. CLL is a malignancy characterized by clonal expansion of CD5 + B-cells that show morphologically mature appearance and accumulate in the blood, bone marrow and secondary lymphoid tissues, resulting in lymphocytosis, bone marrow infiltration, lymphadenopathy and splenomegaly [2].

CLL is a clinically heterogeneous disease, and is divided into two main forms, aggressive and indolent. It is classified by whether the CLL cells express an unmutated (aggressive) or mutated (indolent) immunoglobulin heavy-chain variable region (IGVH) gene, reflecting the stage of normal B cell differentiation. The clinical heterogeneity of CLL reflects differences in the biology of the disease and chromosomal alterations (del13q, del11q, trisomy 12 and del17p), and only in part correlate with aggressive behavior and response to therapy; other events such as somatic mutations, epigenetic changes and non-coding RNAs alterations have been reported to influence the outcome of the disease, but are not used in clinical practice yet.

While much attention has been given to the mutation of protein-coding genes, almost 20% of CLL patients do not show chromosomal aberrations; therefore, it is not surprising that attention has shifted to analysis of the RNA molecules that lack protein-coding potential and are collectively referred to as non-coding RNAs (ncRNAs).

ncRNAs are divided, according to their size, into two groups: small ncRNAs (<200 nt) and long ncRNAs (lncRNAs). Small ncRNAs are mostly constituted by microRNAs (miRNAs), which have received more attention in the last decade, particularly in CLL [3,4]. However, the human genome encodes also numerous lncRNAs, defined as endogenous cellular RNAs of more than 200 nucleotides in length, that lack an open reading frame of significant length. Indeed, lncRNAs are a very heterogeneous group of RNA molecules that cover a broad spectrum of molecular and cellular functions and are deregulated in many human cancers [5,6].

In this review, we will contextualize the role of ncRNAs with respect to the well-known hallmarks of cancer (Figure 1), which are distinctive and have the capabilities to enable tumor growth and metastatic dissemination [7,8].

## 2. Sustaining Proliferative Signaling and Resisting Cell Death

Due to the slow clinical progression of CLL, this disease has been frequently described as the accumulation, rather than proliferation, of malignant B cells, as a result of a deregulated apoptosis. The correct balance between the two biological processes is a key component for CLL characterization. While the accumulation of malignant B cells in the blood, bone marrow and lymphoid organs is caused by an impaired apoptosis, a smaller and actively proliferating population still persists, mostly in lymph nodes (LN) and in bone marrow (BM) [9,10]. miRNAs and lncRNAs are regulating this complex process at various levels (Figure 2), and in the last decade several studies provided evidence to support the role of these RNA types in leukemia and lymphoma progression.

One of the best studied is the miR-17/92 polycistronic miRNA cluster, which is overexpressed in many lymphoid malignancies, including CLL, and comprises miR-17, miR-18a, miR-19a, miR-19b, miR-20a, and miR-92a [11]. A transgenic mouse overexpressing miR-17/92 specifically in B-cells demonstrated that this cluster could act as an oncogene in leukemogenesis; 80% of miR-17/92 transgenic mice developed a B-cell malignancy characterized by expansion of CD19+ B cells [12]. Even if the mechanism that drives miR-17/92 expression remains poorly understood, studies demonstrated that up-regulation of miRNAs belonging to the miR-17-92 cluster in unmutated IGHV CLL is preceded by induction of MYC, providing a link between MYC, B-cell receptor (BCR) activation and miR-transcription in CLL [11,13]. miR-17/92 cluster regulate cell cycle progression and proliferation through many targets, including CDKN1A, CTGF, EGR2, E2Fs, IKAROS, PTEN, STATs, TP53INP1, TRIM8 and ZBTB4 [14]. Among these genes, *ZBTB4* and *TP53INP1* are involved in apoptosis regulation through CDKN1A and TP53 [15], while E2F5 is involved in the G1 arrest [16]. Members of the miR-17/92 cluster, together with the let-7 family of tumor suppressors, miR-26a, and miR-34a have also been found to target Cyclin D-CDK complexes at various levels, allowing for the overcoming of the G1/S checkpoint [17]. Indeed, deregulation of Cyclin D is frequent in CLL and substantially contributes to cell cycle alterations.

A lncRNA with an important functional role in CLL proliferation is MIAT (myocardial infarction associated transcript). This lncRNA located at 22q12.1 with a length of 30,051 bp, was first identified as a vulnerable locus for myocardial infarction [18]. Since its discovery, MIAT was found to be aberrantly expressed in various diseases, such as myocardial infarction, schizophrenia, ischemic stroke, diabetic complications, age-related cataract and cancers [18,19,20]. MIAT is a nuclear-retained lncRNA that shows no association with chromatin and constitutes a unique nuclear structure to evade nuclear export, differently from the lncRNA Xist, which has a crucial structural role for the heterochromatin formation during chromosome X inactivation. MIAT is overexpressed in malignant mature B cells, including established CLL cell lines and cells of non-Hodgkin lymphoma origin as well as primary leukemic cells obtained from CLL patients. MIAT expression correlates with disease aggressiveness, and is directly regulated by *oct4*. MIAT upregulation supports monoclonal malignant B cell proliferation with naïve B-cell phenotype by protecting from apoptosis, and thus, contributing to CLL progression [21].

Programmed cell death is a natural defense mechanism against aberrant proliferation signaling and cancer, and resistance to apoptosis is among the most important hallmarks of cancer, contributing to clonal cell development, tumor growth, and resistance to treatment.

Activation of apoptosis occurs through two main pathways, the intrinsic (mitochondrial-mediated) and the extrinsic (death receptor-mediated), but both types lead to the activation of caspases, and, in turn, to cell death. The intrinsic pathway is tightly regulated by several members of the BCL-2 (B-cell Lymphoma) family, and it is established that CLL is frequently characterized by high expression levels of the antiapoptotic protein BCL-2 [22], where activation is correlated with the loss of miR-15a and miR-16-1, located on Chromosome 13q [23]. Del(13q) is the most common chromosomal aberration in CLL, present in almost 60% of patients, and associated with favorable prognosis. Both miR-15a and miR-16-1 negatively regulate BCL2 at a posttranscriptional level [24], inducing apoptosis in leukemic cell lines [24], and in vivo in xenograft-bearing nude mice [25].

Whereas 13q deletions involving a minimally deleted region (MDR) encompassing DLEU2, DLEU1, miR15a and miR16-1 are a favorable prognostic factor in CLL, larger 13q deletions involving RB1 are associated with poor prognosis in terms of time to first therapy, progression-free survival and overall survival [26,27]. DLEU1 (deleted in leukemia 1) and DLEU2 are two lncRNAs transcribed head to head in the MDR of chromosome 13 [28]. Their homozygous loss has great effects on the regulation and control of normal CD5+ B lymphocytes and their homeostasis, only partially due to loss of miR-15a and miR-16-1, located within intron 4 of DLEU2. Indeed, recent studies showed that DLEU1 and DLEU2 control transcription of their neighboring candidate tumor suppressor genes, which may act as positive regulators of NF-kB activity [29].

In regards to mediation of cell cycle proliferation and induction of apoptosis, a central role has always been ascribed to p53, for which many targets have been identified, especially in CLL. Here, deletions of 17p13.1, the chromosomal region encoding p53, and/or mutations of TP53 are detected in 5–10% of patients at diagnosis. This percentage significantly increases in advanced disease and among those previously treated with chemotherapy, accounting for lack of response and poor prognosis [30]. The frequency of TP53 abnormalities is 10 to 12% in patients at the time of first-line treatment, 40% in patients with fludarabine-refractory CLL, and 50 to 60% in patients with Richter’s syndrome [31]. LincRNA-p21 (long intergenic non-coding RNA p21, located 17 kb upstream of p21) and NEAT1 (nuclear enriched abundant transcript 1) are two p53-regulated lncRNAs induced in response to DNA damage identified in primary CLL cells only when a functional form of p53 is present [32]. Both lncRNAs act in p53-mediated processes, specifically inducing apoptosis and cell-cycle arrest. LincRNA-p21, predominantly acts in trans to activate expression of its neighboring gene, p21. Mechanistically, lincRNA-p21 directly interacts with heterogeneous nuclear ribonucleoprotein K (hnRNP-K), a transcriptional co-factor important in p53-induced p21 transcription. Deregulation of Linc-p21 results in p53 repression, thus inducing apoptosis [33]. Indeed, lincRNA-p21 has been demonstrated to directly correlate with p21 levels in CLL patients [32], even if previous findings proposed a cis-effect of lincRNA-p21 on p21 protein [34]. NEAT1 is also a lncRNA involved in DNA repair response through the p53 pathway [35]. It also plays an essential role as a structural component of paraspeckles, which are protein-rich nuclear organelles built around a specific lncRNA scaffold that influence gene regulation through sequestration of component proteins and RNAs. Hypoxic induction of NEAT1 stimulates paraspeckle formation and leads to accelerated cellular proliferation, improved clonogenic survival, and reduced apoptosis [36]. Indeed, increased NEAT1 levels are associated with enhanced apoptosis in irradiated CLL cells and enhanced chemosensitivity, through the compromised ATR pathway [37].

Interestingly, members of a novel class of small non-coding RNAs were found to be co-deleted with *TP53* gene in 17p deleted CLL patients. Although originally considered as microRNAs, miR-3676 and miR-4521 (now called ts-53 and ts-101) were proven to be tRNA-derived small RNAs (tsRNAs), products of cleavage at specific sites in tRNAs or pre-tRNAs. Their expression does not correspond to precursor tRNA levels [38], demonstrating that tsRNAs are not degradation products, but they play regulatory roles mainly via their association with Argonaute proteins [39]. Both ts-53 and ts-101are down-regulated in all CLL types; in addition, ts-53 is mutated in 1% of CLLs and targets 3’UTR of oncogene TCL1, silencing its gene expression in microRNA-like manner [40]. TCL1 is a key molecule in lymphomagenesis and cancer progression. Even though TCL1 is responsible for development of early B- and T-cells [41] under normal conditions, its deregulation in B-CLL may be a causal event in the pathogenesis of this disease. This is also supported by the observation that TCL1 expression can be regulated by miR-29 and miR-181, which are down-regulated in most of types of leukemia including CLL [40].

HULC (highly upregulated in liver cancer) is a lncRNA located in the short arm of chromosome 6, which acts as an endogenous sponge downregulating a series of miRNAs, including miR-372 and miR-200a-3p [42]. Decreased expression of lncRNA HULC results in increased apoptosis via downregulation of Cyclin D1 and BCL-2 proteins [43], and its expression levels are directly correlated with decreased overall survival (OS) and progression free survival (PFS) and clinical stages, especially in DLBCL patients [43].

On the microRNA side, the evolutionarily conserved microRNA miR-155 is a critical regulator of posttranscriptional gene expression in B cells [44], and is involved to various extents in multiple hallmarks of CLL. miR-155 is transcribed from a region known as the B-cell integration cluster (BIC, or host gene mir-155, miR155HG), which originally was identified as being a frequent integration site for avian leucosis virus. miR155HG is expressed during B-cell differentiation and consists of three exons spanning a 13-kb region at chromosome 21q21. miR155HG shows high expression levels in antigen receptor stimulated B- and T-cells and is found to be increased in CLL, while remaining almost undetectable in healthy samples [45]. Increased expression of miR155HG and miR-155 results from transcription activation by the MYB transcription factor and leads to miR-155-mediated downregulation of several tumor suppressor genes. In this case, the lncRNA miR155HG plays an important role in the regulation of miR-155 which is directly involved in lymphomagenesis or leukemogenesis.

The downregulation of lnc-TOMM7-1, which map to chromosome 7p antisense to the interleukin-6 (IL6) gene, promotes B-cell proliferation. This lncRNA may participate in IL6 transcriptional regulation and therefore may have a pathogenic role, given the potential function of IL6 as an autocrine growth factor in CLL [46].

Another group of non-coding RNAs deregulated in cancer are the circRNAs, which are formed by splicing events where the 5′ and 3′ ends join to form covalently closed continuous loops without polyadenylated tails. In CLL, circ-CBFB, derived from the CBFB transcript, has been reported to be an independent prognosis factor in CLL patients, and is highly upregulated in this cohort compared to healthy donors [47]. In particular, this circRNA is acting as a sponge for miR-607, decreasing its downstream target FZD3, thus contributing to the regulation of the Wnt/β-catenin pathway in CLL, and representing a potential therapeutic target for these patients [47].

Similarly, circ_0132266, is not only downregulated in CLL but can also act as sponge for oncogenic miR-337-3p. This was substantiated by a significant negative correlation between the expression levels of miR-337-3p and circ_0132266 and the dual luciferase reporter assay. Among genes directly modulated by miR-337-3p is the promyelocytic leukemia protein (PML), a tumor suppressor regulating several cellular processes including proliferation, apoptosis, or antiviral response. Low levels of circ_0132266 might via miR-337-3p upregulation lead to PML repression and suppress CLL cells apoptosis and proliferation [48].

## 3. Genomic Instability and Mutations

Genome stability is one of the most important hallmarks of cancer, as it has been found to be detrimental to cell survival [8]. Genomic instability is defined as high mutation frequencies manifested by changes in nucleic acid sequences, chromosome rearrangement, or aneuploidy [49]. Strictly correlated with these mutational events are alterations in DNA damage–repair pathways, which are meant to correct alteration in DNA sequences or prevent them from taking place [50]. Moreover, epigenetic modifications, especially DNA methylation, contribute to maintaining genetic content in multiple ways. In this scenario, several non-coding RNAs have been described as main contributors in maintaining genomic stability and supporting DNA repair proteins.

The methylation levels of the promoter regions of two lncRNAs, AC012065.7, and CRNDE, are inversely correlated with their expression levels. Compared to normal controls, CLL samples displayed hyper-methylation of the CRNDE promoter, and hypo-methylation of the AC012065.7 promoter, and both were found to be associated with poor outcome and shorter overall survival (OS) [51]. Moreover, expression of AC012065.7 and CRNDE were positively correlated with expression of the protein-coding genes *GDF7* and *IRX5* respectively; both are encoded close to the lncRNAs, suggesting *cis*-regulation. GDF7 is known to play an important role in growth, repair, and embryonic development, while IRX5 is involved in apoptosis and cell cycle regulation.

BM742401 is a tumor suppressor lncRNA whose entire sequence and promoter region are embedded in a CpG island and has been identified as inactivated by methylation in CLL [52]. The promoter of this lncRNA is fully methylated in CLL cell lines, while unmethylated in normal controls including normal bone marrow, peripheral blood buffy coats, and CD19+ peripheral B-cells. The functional activity of lncRNA BM742401 mainly relies on epigenetic alteration, and treatment with the hypomethylating agent 5-Aza-2′-deoxycytidine leads to an increase in expression levels of BM742401. Functionally, unmethylation of BM742401 leads to its overexpression in CLL cell lines, resulting in inhibition of cellular proliferation and enhanced apoptosis through caspase-9-dependent intrinsic but not caspase-8-dependent extrinsic apoptosis pathway. These data suggest that lncRNA BM742401 can function as a tumor suppressor in CLL.

Another lncRNA, treRNA, is highly expressed in patients with aggressive CLL, bearing poor prognostic indicators such as unmutated IGHV and high ZAP70 protein expression. In this set of patients, treRNA expression is associated with shorter PFS and OS, and a low expression of treRNA has been proven to be an independent prognostic factor for improved PFS in patients receiving fludarabine plus cyclophosphamide [53]

BGL3 (beta globin locus transcript 3) is a 3.6-kb lncRNA derived from chromosome 11p15.4. BGL3 expression in leukemic cells is negatively regulated by Bcr-Abl through c-Myc–mediated DNA methylation [54]. Conversely, BGL3 regulates Bcr-Abl through sequestering miR-17, miR-93, miR-20a, miR-20b, miR106a, and miR-106b [54], which are known to be PTEN repressors [55].

An exceptional position in DNA damage repair belongs to p53. Mutations in tumor suppressor p53 are therefore connected with poor response or even chemotherapy resistance [31,56]. Up to now, several p53-dependent non-coding RNAs targets have been identified. Beside miR-34a, which has been established as a direct target of p53, this transcription factor also drives expression of miR-182-5p, miR-7-5p and miR-320d/c and two previously mentioned long non-coding RNAs - NEAT1 and lincRNA-p21 [32].

CLL therapy often includes administration of the DNA damaging drugs such as chlorambucil or bendamustine [57], for which p53 functionality and B-cell receptor signaling is crucial. Focusing on miR-34a, this miRNA is most prominently upregulated during DNA damage response (DDR) in CLL. In detail, DDR activation stabilizes p53 which leads to higher expression of miR-34a and subsequent repression of FOXP1, the direct target of this miRNA. As a result, FOXP1 down-modulation limits pro-survival/pro-proliferative signals from the B-cell receptor [58].

## 4. Enabling Replicative Immortality

Telomeres are nucleoprotein structures at the ends of eukaryotic chromosomes, which maintain genome stability by protecting the ends of chromosomes from fusion and degradation [59]. Telomere shortening in duplicating somatic cells eventually leads to the destabilization of the telomere capping structure and to the activation of a DNA damage response. The final outcome of this process is cell replicative senescence, which constitutes a protective barrier against unlimited proliferation. Neoplastic cells acquire unlimited replicative potential by activating a telomere maintenance mechanism that allows them to bypass senescence, one of the main hallmarks of cancer. RNA polymerase II transcribes telomeric sequences and give rise to a class of lncRNAs called TERRA (telomere repeat-containing RNA) [60]. These non-coding RNA molecules are characterized by the presence of telomeric 5′-(UUAGGG)-3 repeats at its 3′ end and are directly involved in the maintenance and regulation of the telomere’s homeostasis. TERRA was described in 2007 as an evolutionarily conserved heterogeneous but specific product that can be generated from almost all telomeres from yeast to mammalian cells [61,62]. TERRA is always transcribed in a centromere-to-telomere orientation and its length depends on the location of the transcription start site (TSS) along the subtelomeric locus, giving rise to TERRA molecules with a length ranging from 100 bp to almost 9 kb [61]. Short TERRA transcripts have the capacity to inhibit the catalytic activity of telomerase reverse transcriptase (TERT) in vitro by associating with the telomerase RNA component (TERC) [63]. TERC is overexpressed in CLL patients bearing the SF3B1 mutation, thus increasing telomerase activity in activated lymphoid cells and is associated with poorer outcome and higher incidence of refractory CLL [64]

## 5. Angiogenesis

The significance of angiogenesis in blood cancers, such as leukemia, has been noted since it plays an important role in both hematopoiesis and leukemogenesis [65]. Even if CLL cells are not dependent on a network of blood vessels to support basic physiological requirements as is true for solid cancers, the bone marrow microenvironment is a fine-tuned balance between T-cells, B-cells, osteocytes, stromal cells, adipocytes, white blood cells, fibroblasts, and endothelial cells regulate the balance between pro-angiogenic factors and inhibitors.

The VEGF family is one of the most widely studied in the field of angiogenesis, and VEGF serum levels in CLL correlate with PFS [66]. The von Hippel-Lindau gene product (pVHL), is expressed at a notably low level in CLL B-cells compared with normal B cells, due to repression by miR-92-1, which is overexpressed in CLL B-cells. This, in turn, stabilizes HIF-1α which transcriptionally activates VEGF, upholding its sustained overexpression and producing abnormal autocrine VEGF secretion in CLL cases [67]. On the same pathway acts miR-155-5p, which, among others, is able to target HIF-1α and influence angiogenesis in the same context [68].

Deregulation of miR-30d was observed in chronic lymphocytic leukemia, where it exerts its pro-angiogenic function, at least in part, by inhibiting MYPT1, which in turn, increased phosphorylation levels of c-JUN and activates the VEGFA-induced signaling cascade [69].

Angiogenesis in CLL is strongly dependent on signals from the tumor microenvironment. Stromal cells can induce c-MYC expression via CD40 signalization which drives expression of miR-9 and miR-17-92 cluster [70]. The miR-17-92 cluster has especially been reported to be significantly overexpressed in CLL and can inhibit TGF-β responses by directly targeting its receptor and/or the key effector SMAD4. This molecular machinery thus down-regulates a wide repertoire of anti-angiogenic factors including clusterin or thrombospondin-1 [71].

Exosomes have been identified as key factors to modulate the formation of new blood vessels and angiogenesis. Exosomes are secreted by both normal and cancer cells and deliver intercellular communication between neighboring cells as well as distant cells by transporting molecules such as proteins, peptides, lipids, miRNAs, and lncRNAs that act as regulators in recipient cells in a paracrine or endocrine manner [72]. Functional involvement in angiogenesis has been described for CLL, where exosomes secreted by leukemic cells contain different functional non-coding molecules such as miR-21, miR-155, miR-146a, miR-148a, let-7g, and miR-451. As a result, CLL exosomes impair the transcriptome of stromal cells and induce the release of cytokines and proangiogenic factors including angiogenine or calmodulin [73].

## 6. Tumor Microenvironment

The utmost importance of tumor microenvironment in CLL is emphasized by several characteristics of this disease, such as the lack of a defined oncogenic driver, the absence of “oncogene addiction”, and the vital importance of BCR signaling, which make CLL cells “addicted to the host.” [74]. As further evidence, in vitro cultures of CLL cells die without support from the microenvironment, such as bone marrow stroma or monocytes. Even in in vivo models, such as in the NSG CLL xenograft model [51], CLL cells depend on a functional microenvironment, particularly the presence of T cells. Indeed, despite most of the circulating CLL cells resting in G0 phase, they maintain the ability to respond to stimuli provided by their interactions with stromal cells and T cells within specific microenvironmental niches.

The microenvironment in CLL has been strongly associated with properties such as proliferation, survival, adhesion, drug resistance, and metastasis [75]. Thus, the intercommunication of CLL cells and their microenvironment is a key factor in understanding the different clinical presentation and pathologic features of CLL.

The most important cellular elements in CLL microenvironment are mesenchymal stromal cells, nurse-like cells and T cells [76] and the interaction between leukemic B cells and cells from microenvironment is mediated by different receptors e.g., chemokine receptors, BCR, CD38, CD40 and TLRs (toll like receptors) [77]. Specifically, CD38 allows CLL cells to interact with CD31, the ligand expressed by both stromal and nurse-like cells. This receptor-ligand interaction leads to activation of ZAP-70, an important enhancer of BCR signaling, whose level has been correlated with increased proliferation and migrative capacity of CLL clones [78] and which can induce transcription of miR-21 via MAPK and STAT3 signaling [79]. It is well documented that miR-21 is overexpressed in malignant B cells [80] and in a mouse model led to development of pre-B-cell-like malignancy showing called the “oncomiR addiction” [81].

Other signals within the CLL microenvironment, such as BAFF or CD154, can affect noncoding RNAs, mainly through activation of NF-κB. In lymphatic nodes, these stimulatory signals from the leukemia cell microenvironment can lead to upregulation of miR-155 and to lower levels of the SHIP1 protein, resulting in higher responsiveness to BCR ligation [82], and therefore proliferation promotion.

The role of the microenvironment is unquestionably important in drug resistance, where soluble factors, cell surface receptors, drug transporters, microvesicles, as well as the direct cell–cell contact take part in the process. TLR key players in host defense from infection have a primary contribution to drug resistance; prestimulation of CLL cells with TLR ligands led to prosurvival signaling and protective effect to fludarabine treatment, marked by upregulation of miR-155-3p and lymphotoxin-α [83].

TGF-β signaling not only plays an important role in mediating proliferation and cell death, but also in regulating miRNA metabolism through a SMAD-dependent mechanism [84]. Previous in vitro and in vivo studies revealed that miR-181a/b family members could be regulated by TGF-β at both transcriptional and processing levels [85]. Consistently, downregulation of miR-181a and miR-181b correlate with TGF-β-SMAD canonical pathway inactivation in CLL cells [86]. On the other hand, overexpression of these miRNAs sensitize CLL cells to fludarabine-mediated cell death by targeting key mediators such as B-cell lymphoma− 2 (BCL− 2), myeloid cell leukemia-1 (MCL-1) and X-linked inhibitor of apoptosis (XIAP) proteins [87].

Previously mentioned members of the miR-15/16 family are known to induce NF-kB pathway involved in increased resistance to anti-cancer therapies. NF-kB mediates signals from crucial pathways in the crosstalk between CLL cells and their protective microenvironment, such as the B-cell receptor (BCR) pathway, the BAFF/BAFF-receptor axis, or the CD40L/CD40 axis [88].

A potential role for miR-125b and miR-532-3p was observed in chronic lymphocytic leukemia patients in response to rituximab monotherapy, since their levels in circulation are inversely correlated with rituximab-induced lymphodepletion. Computational analyses of predicted gene interaction of miR-125b and miR-532-3p revealed a specific targeting of the interleukin-10 pathway and CD20 (MS4A1) family members. Interestingly, not only were miR-125b and miR-532-3p negatively correlated with MS4A1 expression, but they were also positively correlated with MS4A3 and MS4A7 (marker of early myeloid differentiation and myeloid maturity marker respectively) [89]. It has previously been shown that overexpression of miR-125b induce daunorubicin resistance in leukemic cell lines through decreasing expression of G protein-coupled receptor kinase 2 and p53-upregulated modulator of apoptosis [90].

Tumor-associated inflammation is a key factor able to contribute to different hallmarks such as sustained proliferative signaling and angiogenesis. Chronic inflammation supplies cells with growth factors, and newly formed vessels provide nutrients and immune tolerance avoids immune-mediated elimination.

Even in CLL where tumor cells are circulating in the peripheral blood without the presence of a typical tumor microenvironment, the formation of pseudofollicular proliferation centers can be observed. A study observed the activation of leukemic cells by their exposure to antigens and by the interaction with other blood cells [91]. Various molecules take part at different levels in the inflammatory inter-cellular communication such as soluble factors (cytokines) or nano-vesicles (e.g., exosome), which have been shown to carry both proteins and RNAs [92].

Among the intracellular molecules and pathways that drive chronic inflammation, activation of pro-inflammatory intracellular master regulators, such as NF-kB and STAT3, have a leading role, and an additional level of regulation by non-coding RNAs also takes place [93].

PACER (p50-associated COX-2, extragenic RNA) is a lncRNA transcribed from the upstream region of the COX-2 gene, which in turn regulates COX-2 expression by interacting with the repressive p50 subunit of NF-kB, thereby acting as a decoy lncRNA for NF-kB signaling [94]. On the other hand, linc-Cox2, a lincRNA highly induced in macrophages upon TLR ligation, was reported to be an early-primary inflammatory gene controlled by NF-κB signaling. Specifically, after LPS stimulation, lincRNA-Cox2 functioning as a scaffold molecule recruits the SWI/SNF chromatin-remodeling complex, leading to the activation of late primary inflammatory NF-kB -dependent genes [95].

Several miRNAs have been reported to be involved in inflammation/immune response in CLL. For instance, tumor-secreted miR-21 and miR-29a can behave as ligands and bind to the TLR family in immune cells, and thus modulate the cancer–immune system communication responsible for tumor growth and metastasis [96]. Similarly, miR-146a has a central role in the modulation of the TLR pathway by directly targeting TLR4 and signaling proteins such as MyD88 (myeloid differentiation primary response), IRAK-1 (interleukin-1 receptor-associated kinase 1), and TRAF6 (TNF receptor-associated factor 6). TLR activation leads to miR-146a up-regulation in monocytes via the IL-10–mediated STAT3-dependent loop, resulting in significant reduction in the production of proinflammatory cytokines and chemokines, including IL-6, TNF-α, IL-8, CCL3, CCL2, CCL7, and CXCL10 [97]. Up-regulation of miR-146a in CLL monocytes is also critical for endotoxin-induced tolerance, a transient state where immune cells are unable to respond properly to endotoxin challenges and work as a counter response to inflammation. When uncontrolled, this mechanism can lead to a risk for clinical complications such as septic shock or autoimmune diseases [98]. A similar state has been described in cancer, where innate immune cells that infiltrate a tumor develop a “tumor tolerance” leading into decreased production of proinflammatory cytokines, high phagocytic ability, and impaired antigen presentation [99].

CLL-derived exosomes are demonstrated to be important contributors in the maintenance of an inflammatory phenotype. They are actively incorporated by endothelial and mesenchymal stem cells both ex vivo and in vivo and by transferring non-coding RNAs and proteins to the stromal cells, they stimulate proliferation, migration, and secretion of inflammatory cytokines, contributing to a tumor-supportive microenvironment [73].

CLL-derived exosomes rapidly enter into stromal cells (MSCs, Mesenchymal Stem Cells and ECs, Endothelial Cells) through an active process requiring surface proteins such as tetraspanins and integrins and the presence of heparan sulfate proteoglycans on the surface of target cells. The transfer of functionally active miR-146, miR-155 and miR-150 to BM-MSCs trigger an inflammatory reaction, which is a hallmark of many cancers [73].

One of the most upregulated miRNAs in plasma-derived exosomes from CLL patients is miR-19b. This miRNA enhances the proliferation and even the invasiveness of CLL cells by upregulating Ki67 and downregulating TP53, leading to the evolution of therapy resistant CLL into Richter syndrome [100].

lncRNA LINC00461 has been demonstrated to be highly expressed in MSCs-derived exosomes, and stimulates proliferation while its down-regulation decreases ERK1/2 and AKT, expression levels of miR-9, and MEF2C and TMEM161B (transmembrane protein 161B), making it an excellent target for therapeutic applications [101]. Furthermore, LINC00461 targets miR-15a and miR-16, and regulates oncogene Bcl2 expression, contributing to the apoptosis in multiple myeloma and, to a lesser extent, in CLL [101].

Cancer cells could use release of cell-derived EVs also to discard tumor suppressor non-coding RNAs in order to stimulate cancer initiation and progression. An example of this is miR-202-3p, whose selective enrichment was observed in CLL-derived EVs. The exosomal release of this miRNA into the microenvironment helps increases the expression of “suppressor of fused” (Sufu), a negative regulator of Hedgehog signaling and a known target of miR-202-3p [102].

## 7. Conclusions

Non-coding RNAs have been proven to be important regulators of cell cycle homeostasis, and thus, fundamental contributors in cancer development. Alterations in their expression are crucial for acquiring hallmarks of cancer, biological capabilities triggering cancer development and driving disease progression. Depicting the physiological role of ncRNA in normal tissues would help the transition to the use of these molecules as prognostic or diagnostic markers in cancer. Due to the easy access of circulating leukemic cells from blood in CLL, preventive screening of these molecules could offer a lot of advantages in terms of correct therapy selection, prevention of progression or resistance to therapy.

The best example of the key role of non-coding RNAs in the development of new drugs for cancer treatment is the use of venetoclax in patients with high BCL-2 protein levels, correlated with deletions on chromosome 13 and loss of miR-15a and miR-16-1. For the same principle, the development of cirmtuzumab aims to target ROR1, a type-1 tyrosine kinase-like orphan-receptor, also targeted by miR-15a and miR-16-1. Potential examples that could be developed in the near future include the lncRNA BM742401, frequently methylated in CLL [52] and significantly associated with higher lymphocyte counts and advanced Rai stage (≥ stage 2). In vivo experiments demonstrated how the use of hypomethylating agents can lead to promoter demethylation and re-expression of BM742401 in patients with low expression of this molecule.

ncRNAs could be screened for therapy decision; for example, low levels of miR-181a and miR-181b predict therapy resistance, and their upregulation enhances drug sensitivity in primary CLL cell cells through direct targeting of the anti-apoptotic genes BCL-2 and MCL1.

In regards to the direct targeting of lncRNAs in CLL, the potential clinical utility of lncRNA-interference therapy should be explored in in vivo settings and clinical trials in order to develop new and safe therapies for patients.

ncRNAs act through various mechanisms in cancer, and so full comprehension of their mechanisms of action may help in developing new screening and therapeutic approaches. This review highlights the implication of long and short ncRNAs in affecting the most important hallmarks of cancer and suggests how including these biomolecules can improve prognosis and therapy of CLL.

Given their unquestionable biological importance, ncRNAs provide opportunities for anti-cancer therapy development. Nonetheless, only few examples of CLL specific molecules have been functionally characterized and the most important pathways involved in this process are listed in Table 1. Many challenges need to be overcome before clinical application and deeper understanding of ncRNAs functional role in tumor progression still need to be discovered.

## Figures and Tables

**Figure 1 ijms-21-06720-f001:**
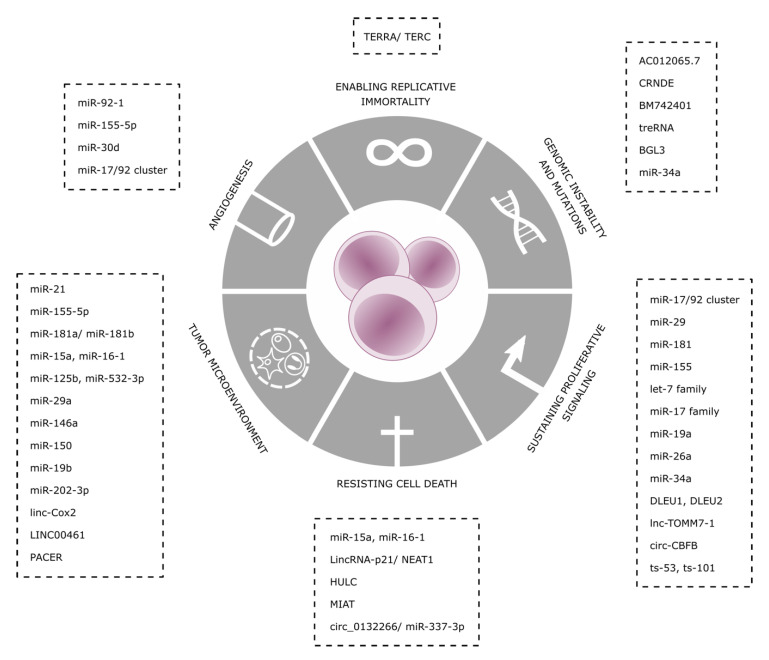
Non-coding RNAs (ncRNAs) as hallmarks of chronic lymphocytic leukemia (CLL).

**Figure 2 ijms-21-06720-f002:**
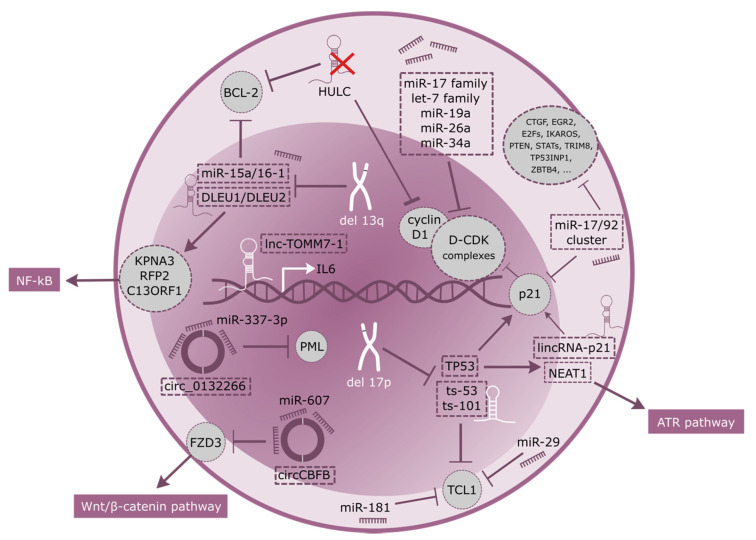
Sustaining proliferative signaling and resisting cell death hallmarks in CLL.

**Table 1 ijms-21-06720-t001:** ncRNAs associated to Hallmarks of Cancer in CLL.

	Non Coding RNA	Mechanism	Levels IN CLL	Ref
**Angiogenesis**	miR-92-1	Represses pVHL and stabilizes HIF-1α, activating VEGF expression	overexpressed	[67]
	miR-155-5p	Targets HIF-1α	High in aggressive disease	[68]
	miR-30d	Inhibits MYPT1, increases phosphorylation levels of c-JUN and activates VEGFA signaling cascade	Reduction	[69]
	miR-17/92 cluster	Targets SMAD4 and inhibits TGF-β responses	overexpression	[71,72]
**Enabling Replicative Immortality**	TERRA/TERC	Inhibition of TERT	Overexpressed in CLL	[60,61,62]
**Genomic Instability**	AC012065.7	positive expression correlation with GDF7	Promoter hypo-methylated	[80]
	CRNDE	Interacts with PRC2 and CoREST to modulate transcriptional repression	Promoter hyper -methylation	[80]
	BM742401	Its expression leads to inhibition of cellular proliferation and enhanced apoptosis through caspase-9-dependent intrinsic but not caspase-8-dependent extrinsic apoptosis pathways	Methylated in CLL	[52]
	treRNA	decreases DNA damage and sensitivity to chemotherapy,	highly expressed, correlates with shorter overall survival (OS)	[53]
	BGL3	regulates the oncogenic expression of BCR-ABL fusion gene through c-Myc mediated signaling	Decrease in CLL	[54]
	miR-34a	Downregulation of FOXP1, limiting BCR signaling	Upregulated during DNA damage response	[58]
**Resisting Cell Death**	miR-15a, miR-16-1	Target and deregulate BCL2	Deleted in 13q- CLL	[24,25]
	lincRNA-p21	Induced by p53Induces p21 through hnRNP-K binding	Increased in p53^WT^ samples	[32,33,34]
	HULC	endogenous sponge downregulating miRNAs, including miR-372 and miR-200a-3p	Upregulated in CLL	[42]
	MIAT	Constitution of a regulatory loop with OCT4	Increased in patients with poor OS	[19,20]
	Circ_0132266	endogenous sponge of hsa-miR-337-3p resulting in a downstream change of target-gene promyelocytic leukemia protein (PML)	decreased in the PBMCs of CLL patients	[48]
**Sustaining Proliferative Signaling**	miR-17/92 cluster	mechanism is poorly understood, but up-regulation of miRNAs belonging to the miR-17-92 cluster is preceded by induction of MYC	Overexpressed in CLL	[11,12,13,14]
	miR-29, miR-181	Targeting TCL1	Downregulated in CLL	[40]
	miR-155	transcriptionally activated by MYB, leads to downregulation of several tumor suppressor genes.	Increased in CLL	[45]
	DLEU1, DLEU2	NF-kB activation. Host of miR-15a/16-1 cluster targeting BCL2	Homozygosis loss	[26,27,29]
	lnc-TOMM7-1	It maps to chromosome 7p antisense to the interleukin-6 (IL6) gene and participate to its transcriptional regulation	downregulation	[46]
	circ_CBFB	acts as a sponge for miR-607, and contributes to the regulation of the Wnt/β-catenin pathway	highly upregulated	[47]
	ts-53, ts-101	tsRNAs that play regulatory roles associating with Argonaute proteins	down-regulated in all CLL types	[40]
**Tumor Microenvironment**	miR-21	Transcriptionally activated by ZAP70, enhancer of BCR signaling	overexpression	[77,78,79]
	miR-155-5p	Decreases SHIP1, resulting in higher responsiveness to BCR ligatio	upregulated	[81]
	miR-181a/miR-181b	regulated by TGF-β, they target BCL− 2, MCL-1 and XIAP	Downregulated	[84,85]
	miR-125-b	negatively regulates MS4A1	levels in circulation are inversely correlated with rituximab-induced lymphodepletion	[87,88]
	miR-19b	Upregulates Ki67 and downregulates TP53	Upregulated in plasma-derived exosomes in CLL	[100]
	miR-202-3p	It targets Sufu, a negative regulator of Hedgehog signaling	selectively enriched in CLL-derived EVs	[102]
	Linc-Cox2	Acts as scaffold molecule recruiting SWI/SNF complex, activating the late primary inflammatory NF-κB-dependent genes	highly induced in macrophages upon TLR ligation	[93]
	LINC00461	Directly correlates with decrease ERK1/2 and AKT activities and expression levels of miR-9, MEF2C and TMEM161B	highly expressed in MSCs-derived exosomes	[101]
	PACER	regulates COX-2 expression and acts as a decoy lncRNA for NF-kB signaling	n.d.	[92]

Abbreviations: pVHL: von-Hippel Lindau tumor suppressor, HIF-1α: Hypoxia-inducible factor 1-alpha, VEGF: Vascular Endothelial Growth Factor, MYPT1: Myosin Phosphatase Target Subunit 1, TGF-β: transforming growth factor beta, TERT: telomerase reverse transcriptase, GDF7: growth differentiation factor 7, PRC2: polycomb repressive complex 2, CoREST: REST corepressor, FOXP1: Forkhead box protein P1, BCR: B cell receptor, BCL2: B-cell lymphoma 2, hnRNP-k: Heterogeneous nuclear ribonucleoprotein K, MCL-1: Induced myeloid leukemia cell differentiation, Cox-2: Cyclooxygenase-2.

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
