# Peer review of "Non-Coding RNAs as Cancer Hallmarks in Chronic Lymphocytic Leukemia"

_ijms, 2020, doi:10.3390/ijms21186720_

Round 1
Reviewer 1 Report
I have read with interest the review by Fabris et al. from G. Calin group focusing on ncRNAs as cancer hallmarks in CLL. In general, review contains uptodate information, however its presentation needs to be improved. The review needs profound English editing as missuse of English makes it hard to read. Specific recommendations and suggestions how to improve the review are listed bellow.
Major concerns:
- manuscript needs deep English editing by native Ensligh speaker as many typos occur thorough the text
- unify the use of abbreviations thorought the text, abbreviations shall be explained on the first time they occur in the manuscript, sometimes you explain the same abbreviation even three times (e.g. for TLR)
- the section "Sustaining Proliferative Signaling and Resisting cell death" is written more like a "list" than a "narrative review", please, co through the section and try to connect the individual paragraphs into the "narration", also some miRNAs repeat in different paragraphs (miR17-92 cluser, miR-155 etc.) - try to interconnect the paragraph by reffering to that was already written or what is about to come
- for the section "Sustaining Proliferative Signaling and Resisting cell death" I suggest that authors create a summarizing figure to individual miRNAs, lncRNA, tdRNA, circRNAs and their putative targets to provide the reader with clearer overview of the situation; this cancer hallmark is essential for CLL thus i do believe that summarizing figure would be benefitial
- there is no Table 1 in the manuscirpt provided for review
Minor concerns:
- in abstract, Chronic Lymhocytic Leukemia can be abbreviated to CLL and during the second occurence in the last sentence the abbrevation may be used
- line 27 - please specify the year (2020) in the sentence "This year (2020)..." - as the review will be read and accessed also in the following years
- line 29: please specify "bone marrow" instead of only "marrow"
- please rephrase: "Being a clinically heterogeneous disease, CLL is mainly divided into two forms, aggressive and indolent, classified by whether the CLL cells express unmutated (aggressive) or mutated (indolent) immunoglobulin heavy‐chain variable region (IGHV) gene , reflecting the stage of normal B cell differentiation from which they originate."
- please explain line 38 - "but they are not actionable yet " - do you mean that they are not used in the clinical practice? or that they are not easily determined?
- please rephrase and edit the sentences on lines 56-62 - divide them into more sentences, current phrasing is too complicated and sentence does not have sense (you are stressing out that apoptosis is crutial two times but there is no need for that; e.g. "...can be attributed more to an impaired apoptosis..." - when you use "more" then it shall follow with "than", e.g. "than excessive proliferation")
- line 66-67: put the miRNAs belongin to miR-17-92 cluster into numerical order
- line 83: "The intrinsic pathway is composed by several BCL" please, use different word to "compose", e.g. "The intrinsic pathway abberation is a common sign in several BCL..."
- line 90-91 plase specifiy the difference between "13q deletion" and "large deletion of 13q" as the readers may not be so used in the topic
- line 114: please define what are paraspeckles
- line 143: please define the roles of Xist (as you are comparing MIAT with it but the readers may not be aware of its roles)
- line 394 - why do you suddenly write about multiple myelomu in the review focused on CLL?
Typos:
- line 26: dual spacing between words "representing more"; in general there is a huge occurence of dual spacing thorough the manuscript - please correct them all
- line 36: "correlates" shall be "correlate"
- line 45: change "in particular" to "particularly"
- line 85: "mirR‐15a" shall be "miR-15a"
- line 273 - please edit refference (Willimott, 2012)
Author Response
I have read with interest the review by Fabris et al. from G. Calin group focusing on ncRNAs as cancer hallmarks in CLL. In general, review contains up-to-date information, however its presentation needs to be improved. The review needs profound English editing as misuse of English makes it hard to read.
We would like to thank the reviewer for the positive comment. We revised the English in our manuscript accordingly, to make it easier for the reader.
Unify the use of abbreviations thorough the text, abbreviations shall be explained on the first time they occur in the manuscript, sometimes you explain the same abbreviation even three times (e.g. for TLR)
Thanks to the reviewer, we unified the abbreviation and we made sure they were explained on the first time they appear in the text
The section "Sustaining Proliferative Signaling and Resisting cell death" is written more like a "list" than a "narrative review", please, co through the section and try to connect the individual paragraphs into the "narration", also some miRNAs repeat in different paragraphs (miR17-92 cluster, miR-155 etc.) - try to interconnect the paragraph by referring to that was already written or what is about to come
We modified the paragraph trying to connect the different paragraphs, as suggested.
For the section "Sustaining Proliferative Signaling and Resisting cell death" I suggest that authors create a summarizing figure to individual miRNAs, lncRNA, tdRNA, circRNAs and their putative targets to provide the reader with clearer overview of the situation; this cancer hallmark is essential for CLL thus i do believe that summarizing figure would be benefitial
We agree with the reviewer, and we included an additional figure with an overview of these hallmarks.
There is no Table 1 in the manuscript provided for review.
We thank the reviewer for letting us know this, the table is now included in the main text.
Minor concerns:
-In abstract, Chronic Lymphocytic Leukemia can be abbreviated to CLL and during the second occurrence in the last sentence the abbrevation may be used
-Line 27 - please specify the year (2020) in the sentence "This year (2020)..." - as the review will be read and accessed also in the following years
-Line 29: please specify "bone marrow" instead of only "marrow"
-please rephrase: "Being a clinically heterogeneous disease, CLL is mainly divided into two forms, aggressive and indolent, classified by whether the CLL cells express unmutated (aggressive) or mutated (indolent) immunoglobulin heavy‐chain variable region (IGHV) gene , reflecting the stage of normal B cell differentiation from which they originate."
-Please explain line 38 - "but they are not actionable yet " - do you mean that they are not used in the clinical practice? or that they are not easily determined?
-Please rephrase and edit the sentences on lines 56-62 - divide them into more sentences, current phrasing is too complicated and sentence does not have sense (you are stressing out that apoptosis is crucial two times but there is no need for that; e.g. "...can be attributed more to an impaired apoptosis..." - when you use "more" then it shall follow with "than", e.g. "than excessive proliferation")
-Line 66-67: put the miRNAs belonging to miR-17-92 cluster into numerical order
-Line 83: "The intrinsic pathway is composed by several BCL" please, use different word to "compose", e.g. "The intrinsic pathway aberration is a common sign in several BCL..."
-Line 90-91 please specify the difference between "13q deletion" and "large deletion of 13q" as the readers may not be so used in the topic
-Line 114: please define what are paraspeckles
-Line 143: please define the roles of Xist (as you are comparing MIAT with it but the readers may not be aware of its roles)
-Line 394 - why do you suddenly write about multiple myeloma in the review focused on CLL?
We would like to thank again the reviewer for the suggestions, we modified the text accordingly. We rephrased the unclear sentences and added missing definitions.
Typos:
- line 26: dual spacing between words "representing more"; in general there is a huge occurence of dual spacing thorough the manuscript - please correct them all
- line 36: "correlates" shall be "correlate"
- line 45: change "in particular" to "particularly"
- line 85: "mirR‐15a" shall be "miR-15a"
- line 273 - please edit refference (Willimott, 2012)
We would like to thank the reviewer for helping us correcting the typos. All of the have been modified.
Reviewer 2 Report
The manuscript makes it clear why the creation of this review was important. The details are then presented clearly and carefully. The information gathered should be helpful for researchers in this area and is also interesting for readers who are not directly involved in this area. Only are small write errors noticed - spaces between sentences and individual words are sometimes missing. However, this can be corrected easily.
Author Response
The manuscript makes it clear why the creation of this review was important. The details are then presented clearly and carefully. The information gathered should be helpful for researchers in this area and is also interesting for readers who are not directly involved in this area. Only are small write errors noticed - spaces between sentences and individual words are sometimes missing. However, this can be corrected easily.
We would like to thank the reviewer for the positive comments regarding our work. We corrected all the errors in the text.
Reviewer 3 Report
In this review the authors summarize the recent advances in our knowledge about CLL and miRNA. They subdivide the work in paragraphs on the basis of mechanisms which are well known to be de-regulated in CLL and list those nc-RNAs whose modulated expression infers on them.
The work is potentially interesting, but it is a quite heavy reading, where all nc-RNAs and their effects are listed.
Main criticisms.
- The authors present the literature essentially as a list of nc-RNAs and the effect of their deregulation in CLL. To avoid this aseptic perception, the authors should organize all information regarding nc-RNAs in CLL and their aberrant expression in CLL in a table, which should also summarize the main functions of these RNAs in wild-type cells.
- The conclusions must not be written as a summary of the work. They should instead include take out and spread a take-home-message. I suggest transferring the information regarding how nc-RNA expression affects CLL treatments, that authors report here and there in the text, here in this paragraph bringing this concept to light.
Additional suggestions.
- To my knowledge, ncRNA stands for “non-coding RNA”, not for “non-protein coding RNA”, as the authors wrote. Please check.
- Lines 90-97. The authors might rewrite the sentences regarding the role of DLEU1 and 2 on NF-kB activity to improve comprehension
- Line 143. The authors refer to Xist without including any information about it. They must at least briefly explain what Xist is and what is its role in heterochromatinization of the X chromosome.
- Lines 329-330. NF-kB: B is NOT a Greek letter.
- Lines 185-190 and 220-221. References should be added to the text.
- Line 273. The reference “Willimott 2009” is not formatted.
- I cannot find Table 1 that authors mention in the conclusions. Do they refer to figure 1?
Author Response
In this review the authors summarize the recent advances in our knowledge about CLL and miRNA. They subdivide the work in paragraphs on the basis of mechanisms which are well known to be de-regulated in CLL and list those nc-RNAs whose modulated expression infers on them.
The work is potentially interesting, but it is a quite heavy reading, where all nc-RNAs and their effects are listed.
We would like to thank the reviewer for the positive feedback regarding our work. We agree with the reviewer, we tried to modify English and sentences to make it easier for the reader.
- The authors present the literature essentially as a list of nc-RNAs and the effect of their deregulation in CLL. To avoid this aseptic perception, the authors should organize all information regarding nc-RNAs in CLL and their aberrant expression in CLL in a table, which should also summarize the main functions of these RNAs in wild-type cells.
We would like to thank the reviewer for the suggestion, the table is now included in the main text.
- The conclusions must not be written as a summary of the work. They should instead include take out and spread a take-home-message. I suggest transferring the information regarding how nc-RNA expression affects CLL treatments, that authors report here and there in the text, here in this paragraph bringing this concept to light.
We modified the conclusion paragraph adding insights regarding potential treatments involving ncRNAs.
We also carefully considered all the additional suggestions from the reviewer, and modified the text accordingly.
Round 2
Reviewer 1 Report
I have read with interrested the revised version of the manuscript, that is now much more clearly written and not misleading any more. Figure 2 is outstanding! All my comments were adressed adequatelly.